# Drug-Induced Myopathies: A Comprehensive Review and Update

**DOI:** 10.3390/biomedicines12050987

**Published:** 2024-04-30

**Authors:** Sebastian Miernik, Agata Matusiewicz, Marzena Olesińska

**Affiliations:** Department of Connective Tissue Diseases, National Institute of Geriatrics, Rheumatology and Rehabilitation, 02-637 Warsaw, Poland; sebastian.miernik@spartanska.pl (S.M.); marzena.olesinska@spartanska.pl (M.O.)

**Keywords:** drug-induced myopathies, myositis, checkpoint inhibitors, glucocorticosteroids, myalgia

## Abstract

Drug-induced myopathies are a common cause of muscle pain, and the range of drugs that can cause muscle side effects is constantly expanding. In this article, the authors comprehensively discuss the diagnostic and therapeutic process in patients with myalgia, and present the spectrum of drug-induced myopathies. The review provides a detailed analysis of the literature on the incidence of myopathy during treatment with hypolipemic drugs, beta-blockers, amiodarone, colchicine, glucocorticosteroids, antimalarials, cyclosporine, zidovudine, and checkpoint inhibitors, a group of drugs increasingly used in the treatment of malignancies. The article considers the clinical course of the different types of myopathies, their pathogenesis, histopathological features, and treatment methods of these disorders. The aim of this paper is to gather from the latest available literature up-to-date information on the course, pathophysiology, and therapeutic options of drug-induced myopathies, to systematize the knowledge of drug-induced myopathies and to draw the attention of internists to the fact that these clinical issues are an important therapeutic problem.

## 1. Introduction

Myalgia (muscle pain) is a frequent symptom experienced by most of the population at certain times of life. Predisposing factors include female gender, non-Caucasian race, active nicotinism, older age, low as well as excessive physical activity, lower social status and educational level, and concomitant depression [1]. The diagnostic algorithm for myalgia is presented in Table 1. 

Causes of myalgia differ in accordance with age, gender, past medical history, and many other factors. The main causes of muscle pain with examples are presented in Figure 1. 

Among the frequent factors causing myalgia are myopathies. They are a heterogeneous group of disorders primarily affecting the skeletal muscle structure, metabolism, or channel function. They usually present with muscle pain or muscle weakness interfering with daily life activities. An important cause of myopathies that should always be kept in mind is drug-induced myopathies. The range of drug-related muscle symptoms is wide, including myalgia, cramps, muscle weakness, exercise intolerance, or even rhabdomyolysis and myositis. With the introduction of new medications and their use by a great number of people around the world, the range of recognized myopathic effects associated with these drugs has grown. Continuous observation and research also allow us to broaden our knowledge of the already known drug-induced myopathies. While severe forms of drug-induced myopathy are uncommon, milder forms are probably more frequent than is appreciated. The cause-and-effect relationship between the use of a drug and the onset of symptoms is often difficult to establish. The drug-induced causes of myopathy warrant special emphasis because they are often overlooked, resulting in misdiagnosis and improper care. Recognition of this condition is essential early in its course to determine when to discontinue therapy and potentially prevent irreversible muscle damage. Drug-induced myopathies are of great clinical importance, because although they usually have a mild course, in some cases they can cause severe symptoms and persistent complications and significantly reduce the quality of life. In the following chapters, the authors characterize the types of drugs frequently associated with adverse muscular effects and present a new group of medications whose immunologic adverse effects are now emerging entities.

## 2. Objectives

The authors aimed to analyze the literature available to date on drug-induced myopathies and to characterize the most common ones concerning their clinical course and treatment. We aimed to present this topic comprehensively to attract the attention of clinicians and to raise awareness of these clinical features.

## 3. Methods

The PubMed database was searched for articles using the keywords “drug-induced myopathies” to find relevant articles on the subject available up to 1 January 2024. Among the inclusion criteria were the restriction to case reports, case series, and review articles, as well as articles available in English, and the availability of full text. The exclusion criteria consisted of duplicate articles or abstracts only, and articles published in languages other than English. The authors pinpointed a huge number of articles, among which they selected those presenting up-to-date knowledge to find the main groups of drugs causing myopathy. After that, the search was broadened to include keywords for each group of drugs, for example, “statins associated myopathy”. In each of the types of medications characterized, the most recent and most relevant publications were primarily used. Articles listed as sources for the papers were also utilized to deepen the knowledge of the topic.

## 4. Drugs Causing Myopathies

### 4.1. Hypolipemic Medicines

The medications most associated with muscle side effects are β-Hydroxy β-methylglutaryl-coenzyme A (HMG-CoA) reductase inhibitors (statins). These drugs have a strong potential to reduce the risk of cardiovascular disease, the most common cause of death in developed societies. Unfortunately, neuromuscular symptoms are the most common cause of withdrawal of these medications [2]. According to a study by Abed et al., 27% of patients treated with statins reported muscle-related adverse effects [3]. In other studies conducted so far, the incidence of myopathy in patients using these drugs has been 5–20%. Difficulties in determining the incidence of this complication are in part due to the various definitions of myopathy adopted by different investigators [4,5,6,7].

A higher incidence of myotoxicity is seen with the use of lipophilic statins compared to hydrophilic ones and with higher doses of the drug [8]. In the previously mentioned study by Abed et al., muscular side effects occurred most frequently in the patients taking simvastatin, whereas they occurred least frequently in those taking fluvastatin and rosuvastatin [3]. In contrast, according to a study by Ramkumar et al., the use of rosuvastatin and atorvastatin may reduce the incidence of myopathy by increasing the interval between drug doses thanks to their long half-life [9]. According to the FDA (Federal Drug Administration) Adverse Effects Reporting System (AERS), the risk of serious adverse effects such as rhabdomyolysis is several times greater with the use of lovastatin, simvastatin, and atorvastatin compared to pravastatin, rosuvastatin, or fluvastatin. Concomitant use of fibrates increases the risk by 10 times [10].

The main risk factors for muscle-related side effects are elderly age, female gender, Asian origin, concomitant use of drugs that affect statin metabolism (e.g., macrolides, warfarin, cyclosporin, azole antifungals, diltiazem, or fibrates), increased physical activity, hepatopathy, chronic renal disease, hypothyroidism, vitamin D deficiency, or metabolic syndrome [2,8,11].

Interesting tools for assessing the likelihood of whether patient-reported muscle symptoms are related to taking statins are The Statin-Associated Muscle Symptom Clinical Index (SAMS-CI) and Statin Experience Assessment Questionnaire (SEAQ) [12,13].

The most common SAMSs (statin-associated muscle symptoms) are myalgia and muscle weakness, primarily affecting the proximal muscles; these usually occur at the beginning of therapy but can occur even after many years of treatment [14]. According to a study by Turner et al., muscle symptoms affect mostly the lower extremities, but the shoulder rim and axial muscles can also be involved [15].

A seven-step classification of statin-related myotoxicity (SRM) has been proposed, ranging from SRM0 (asymptomatic creatine kinase elevation), through SRM5 (rhabdomyolysis) to SRM6 (immune-mediated necrotizing myopathy) [16].

Different hypotheses have been proposed to understand the causes of myopathy induced by statin. The pathomechanism of myocyte damage using statin drugs is not fully understood, but it is currently thought to be related to mitochondrial damage. Decreased synthesis of high-energy compounds (e.g., adenosine triphosphate) and impaired electron transfer along the respiratory chain complex have been found relevant. The role of coenzyme Q10 deficiency has also been raised but studies conducted with supplementation of this protein have presented inconclusive results so far [17,18]. In the vast majority of patients, mild muscle symptoms resolve after discontinuation of drugs. It has been also described that, due to statin effects, the depletion of cholesterol in muscle cell membranes may have an important role in causing myopathy, triggering sarcolemma destabilization and altering ion balance [19]. In addition, statins have been found to inhibit the AKT/mTOR signaling pathway, which is involved in muscle growth during development and regeneration [20].

A complication of statin therapy that researchers are now focusing on is necrotizing myositis. This is a rare, specific subset of statin-induced myotoxicity. A feature that distinguishes this condition from other types of statin-related myopathies is the presence of anti-HMG-CoA reductase (HMGCR) antibodies [21]. In these patients, statin treatment seems to be related to an increase in HMGCR expression, which together with a particular genetic predisposition to autoimmune diseases can induce the production of anti-HMGCR. The main symptom is a reduction in the strength of the muscles of the pelvic girdle, with a particular predisposition to the thigh and gluteal muscles. Fatigue and myalgia are present at onset in about 20–60% of patients, and dysphagia affects about 16-30% of people. In some cases, trunk muscle weakness is the only symptom of the disease. Extramuscular symptoms, including skin lesions, arthritis, or Raynaud’s phenomenon, are not frequent [22,23].

A strong association between the presence of the HLADRB1*11:01 allele and statin-related immune-mediated necrotizing myopathy (IMNM) has been reported, reaching up to 70%. The diagnosis of IMNM is based on electromyography (EMG), MRI, muscle biopsy, and the presence of characteristic antibodies [24]. The elevation of creatine kinase (CK) activity may vary from 10 to 100 times above normal [23].

It is important to underline that statin-associated IMNM has to be recognized as soon as possible because it may require a different therapeutic regimen than other inflammatory myopathies. Every patient with persistent muscle weakness and hyperCKemia despite discontinuation of statins should be evaluated for IMNM by determination of anti-HMGCR antibody levels.

### 4.2. Cardiological Drugs

Beta-blockers stand out among the cardiovascular drugs that can cause muscular symptoms. Most of them have muscle cramps or muscle weakness listed among their undesirable effects. However, there are few recent publications on myopathy or myalgia associated with the use of these drugs. Subsequent use of sotalol and then propranolol has been linked to a case of myopathy presenting with proximal muscle weakness and increased CK levels. The muscle biopsy was normal, and the symptoms withdrew after discontinuation of the drug [25]. There are also publications about labetalol-induced myopathy presenting as myalgia and proximal muscle weakness with elevated creatine kinase activity and myopathic recordings on EMG associated with its use. The histopathological picture was consistent with drug-induced myopathy. After discontinuing the drug, the symptoms withdrew and the kinase activity normalized [26,27].

However, according to the available literature, the use of B-blockers with intrinsic sympathomimetic activity (ISA), like pindolol and carteolol, carries a higher risk of muscle cramps and a slight increase in creatine kinase activity [28]. To date, there are no publications on the more serious muscular complications associated with the use of these drugs.

Amiodarone is another drug, whose use has been linked to the occurrence of myopathy, often associated with neuropathy. This complication is rare, but few cases are reported in the literature. The most common clinical presentation is predominantly proximal muscle weakness and markedly high creatine kinase activity. On muscle biopsy, a pattern of severe vacuolar myopathy with muscle fiber deterioration is present. The typical histopathological finding is fatty deposits in the muscle fibers [29,30,31]. Two cases of acute necrotizing myositis associated with amiodarone have been described [32,33].

### 4.3. Colchicine

Myopathy associated with colchicine use is a well-known clinical feature. Its symptoms resemble polymyositis with decreased proximal muscle strength, myalgia, and increased CK activity. Associated neuropathy is often present. There is a higher risk of colchicine-induced myopathy in patients with chronic kidney disease. Histopathologically, a picture of lysosomal, vacuolar myopathy is present, with few necrotic fibers. The only type I muscle fibers are vacuolized. Electron microscopy shows perinuclear aggregates of fibrillar material, and spheromembranous bodies are present. This resembles the microscopic pattern present in other eukaryotic cells treated with antimicrotubule agents. Withdrawal of the drug results in the resolution of symptoms and normalization of muscle laboratory parameters [34,35].

### 4.4. Steroids

There are two types of steroid-induced myopathy.

The first is uncommon and typically concerns critically ill patients, and has the presentation of acute quadriplegic myopathy (AQM). It occurs mainly in critically ill patients with status asthmaticus or other disorders requiring intensive care, and often in those treated with nondepolarizing muscle-blocking agents (NMBAs) to enable mechanical ventilation. Although AQM mainly affects patients receiving large doses of intravenous corticosteroids, relatively low doses, and even a single dose, have been implicated in case reports. Various routes of administration have also been described, including not only intravenous, but also oral, intramuscular, and epidural. The exact mechanism of the muscle weakness is unclear but may be related to altered electrical excitability of muscle fibers, loss of thick filaments, and/or inhibition of protein synthesis and hypokalemia. All of these pathways are believed to increase the rate of muscle catabolism and result in loss of muscle movement [36]. No single test is diagnostic for this condition. Muscle enzymes are often normal; electromyography may be either normal or reveal nothing but decreased amplitude of muscle action potential [37,38]. Histopathologically, it has the appearance of acute necrotizing myopathy. The course is often fulminant, worsening the prognosis, and often results in respiratory distress when attempts to discontinue ventilatory therapy are made [37,39]. There is no specific treatment available for this condition. The most consistent finding in the literature review is the fact that myopathy resolves with discontinuation of steroid therapy.

The second, more frequent type, is associated with chronic use of glucocorticosteroids (GCS). Mostly, the onset of myopathy is insidious, and the first manifestations are usually a decrease in muscle strength, and difficulty climbing stairs or rising from a sitting position, but less often myalgia. Symptoms usually initially affect the proximal muscles of the lower limbs symmetrically, and then symptoms from the shoulder girdle may also join. A cushingoid body appearance is predominantly present. The risk of myopathy is higher with doses higher than 40 mg/day of prednisone equivalent. Creatine kinase activity is mostly normal. On muscle biopsy, the lesions are uncharacteristic, with no signs of inflammation or necrosis. Type II muscle fiber atrophy is present. Treatment consists of discontinuation, dose reduction, or conversion of GCS to non-fluorinated molecules (prednisolone, hydrocortisone) rather than fluorinated molecules (triamcinolone, dexamethasone) [40,41,42]. Rehabilitation regimens and physical activity help manage symptoms [43].

It is often difficult to distinguish between an exacerbation of inflammatory muscle diseases and myopathy associated with chronic steroid use in patients suffering from these entities. The main difference is that creatine kinase activity is usually normal in the case of an association with GCS use, but elevated in the case of disease flare. Often the only way to know the cause of myopathy is through a clinical trial, reducing the steroid dose, and observing symptoms [44].

### 4.5. Antimalarials

Myopathy associated with the use of antimalarial drugs, both chloroquine and hydroxychloroquine, is a known complication of this treatment. Usually, its symptoms are mild, characterized mainly by a decrease in the strength of the proximal muscles of the upper and lower limbs. However, dysphagia, also severe, may be present in several cases. Higher cumulative doses and longer exposure to the drug are associated with more severe symptoms and a higher risk of myocardial involvement and dysphagia [45]. It is usually accompanied by elevated lactate dehydrogenase and creatine kinase activity in laboratory analyses.

A meta-analysis published in 2021 shows that, in studies to date, the incidence of antimalarial myopathy has varied significantly depending on the assessment method used and is difficult to determine conclusively [46]. According to the prospective study published by Casado et al., the incidence of antimalarial myopathy (defined as the presence of microscopic changes with elevated muscle enzyme activity) was 12.6% among patients treated at their rheumatology unit. Of these, about half presented with decreased muscle strength, mostly of mild to moderate severity. After discontinuation of antimalarial drugs, symptoms significantly decreased or disappeared and kinase activity showed a decreasing trend in all patients, which was also confirmed in other studies [47].

The likely mechanism of toxicity is an accumulation of the drug in lysosomes. It results in a pathological increase in environmental alkalinity in these organelles. At pH values above 7.4, the activity of enzymes that metabolize lipids and glycogen is reduced, leading to their storage in vacuoles. The microscopic image of myopathy associated with the use of antimalarial drugs is the formation of large vacuoles and curvilinear bodies with an accumulation of glycogen and myeloid bodies. Curvilinear bodies are characteristic of this kind of myopathy and are present between myofibrils and in the area surrounding the cell nucleus. They persist over time, even after the complete resolution of the symptoms [48]. This pattern is infrequently seen and may be similar only in the case of one rare lipid storage disease-ceroid lipofuscinosis [49].

Hydroxychloroquine, compared to chloroquine, causes less severe structural changes [48].

### 4.6. Cyclosporine A

Cyclosporine can cause myopathy manifested by myalgia, muscle weakness, and an increase in creatine kinase activity [50]. From post-marketing surveillance data, the incidence of cyclosporine-associated myopathy is 0.17% [51]. The pathogenesis of the muscle disorders is believed to be mitochondrial dysfunction resulting from the therapy, but to date, no conclusive work has been published to confirm this theory [52]. The histopathological picture is varied, with no distinctive features. Discontinuation of cyclosporine leads to withdrawal of the symptoms.

### 4.7. Zidovudine

Zidovudine is an inhibitor of human immunodeficiency virus (HIV) reverse transcriptase, widely used to treat retroviral infection. Long-term therapy with this pharmaceutical can lead to mitochondrial myopathy, whose main symptoms are proximal muscle weakness, myalgia, and fatigue. Creatine kinase activity is usually elevated. The incidence of myopathy in patients treated with zidovudine is estimated to be around 17%. The symptoms usually resolve about 4 weeks after cessation of treatment [53]. Histopathology images show damage to the mitochondria with an accumulation of lipids and glycogen subsarcolemmal. Ragged, red fibers (“Z fibers”) are present. The reduction in myofilament number is noticeable, particularly in type 1 fibers [54,55]. mtDNA replication is impaired and cytochrome c oxidase deficiency is present in this myopathy [56,57].

### 4.8. D-Penicillamine

D-penicillamine was a drug commonly used in the treatment of rheumatoid arthritis in the past and is currently used to treat Wilson’s disease. The limitation of its usage is due, among other things, to the reported side effects, the notable ones being the risk of developing an inflammatory myopathy resembling polymyositis or dermatomyositis. The main symptoms are a decrease in proximal muscle strength with a significant increase in CK activity. A rash characteristic of dermatomyositis may be present. A possible complication is swallowing disorders. The etiopathogenesis remains unknown. The histopathological image shows an inflammatory myopathy with necrosis of muscle fibers. Most symptoms withdraw after treatment with corticosteroids. In some patients, penicillamine was continued or reintroduced without relapse of the symptoms [58,59,60].

### 4.9. Checkpoint Inhibitors

Immune checkpoint inhibitors (ICIs) are modern drugs used to treat numerous neoplasms, including malignant melanoma, non-small cell lung cancer, renal cell carcinoma, colorectal cancer, bladder cancer, neuroendocrine skin cancer, and Hodgkin lymphoma. Currently used molecules from this group act against cytotoxic T-lymphocyte-associated protein (CTLA-4; ipilimumab) and programmed death 1 pathway (PD-1; pembrolizumab, nivolumab, and PD-Ligand1 (PD-L1); durvalumab, atezolizumab, and avelumab) [61]. Their novel mechanism of action by harnessing the immune system is promising but also carries several potential threats. The attenuation of T cell inhibitory mechanisms by ICIs leads to hyperactivation of the immune system. This is associated with a variety of adverse events characterized by inflammation. Target sites of these adverse events, usually termed immune-related adverse events (ir-AEs) can include the gastrointestinal tract, endocrine glands, liver, and skin, but cardiovascular, neurological, pulmonary, and rheumatic ir-AEs are also reported. One of the common side effects reported in clinical trials of these drugs is muscle pain. It was reported in about 4–5% of patients receiving anti-PD-1 therapy [62,63].

A more threatening clinical feature that may present initially with myalgia is myositis. According to recent publications, the incidence of biopsy-proven myositis in patients treated with PD-1 inhibitors can reach 0.8%. An increase in the frequency of reporting this feature was noted, perhaps due to greater awareness among physicians [64].

Usually, the symptoms occur early after initiation of the therapy, within the first 30 days of treatment with rapid progression. They may have an appearance of polymyalgia-like syndrome [65]. The first symptom of ICI-associated myositis in most patients is myalgia, occurring in 80% of cases. Very often it precedes the appearance of muscle weakness. This is a remarkable observation since myalgia is not a symptom highly characteristic of another autoimmune myositis. The muscle weakness, when present, is most often symmetrical and involves the pelvic girdle and extensor muscles of the neck. Rarely, dyspnea, fatigue, fever, and chest pain are present. To date, isolated cases of patients with a skin rash characteristic of dermatomyositis have been reported. Associated interstitial lung disease is not common. Myocarditis is a frequent complication of ICI-associated myositis, occurring in 16–32% of patients and necessitating prompt therapy [65,66,67,68]. Muscle symptoms during ICI treatment may also manifest as myasthenia gravis. It occurs in the early phase after ICI treatment with rapid deterioration. Asymmetric drooping of the eyelid or weakness of the oculomotor muscles causing double vision is a common symptom, occurring in about half of the patients. The positivity rate of autoantibodies to the acetylcholine receptor (AChR) is estimated to be about 50% in patients with ICI-induced myasthenia gravis [68]. In contrast, muscle-specific kinase (anti-MuSK) antibodies are generally not detectable. Of note, there was a strong association between ICI-induced myositis and the co-occurrence of myasthenia in 11.9% of the cases, resulting in increased mortality [69]. Therefore, if the patient present with symptoms more pronounced than muscle pain and limb weakness, AChR levels and electrodiagnostic testing including repetitive nerve stimulation and single-fiber EMG should be strongly considered.

Myositis associated with the use of these drugs has a more abrupt onset, no fluctuation of symptoms, or increased fatigue compared to the group with idiopathic inflammatory myopathies [70].

In diagnosis, a significant role is played by the determination of creatine kinase activity, which is elevated in most cases of muscle inflammation (median of 2650 IU/L, ranging from 335 to 20,270 IU/L), but is in a normal range in people with isolated myalgia. Predominantly, myositis-associated antibodies are negative. In one study, the incidence of these antibodies was up to 29% of the patients [68,70].

Muscle biopsy specimens have a characteristic histopathological picture, not seen in other inflammatory myopathies. The most common microscopic picture is that of severe necrotizing myositis, with local areas of necrosis of muscle fibers and the presence of T-lymphocyte and macrophage infiltrates [65].

Treatment depends on the assessment of the severity of the side effects. It is important to evaluate the benefit–risk ratio of discontinuing the drug in the patient considering the progression of the underlying disease and potential muscle damage resulting from active myositis. In its mild course, with myalgia and low or absent CPK elevation, it responds well to moderate doses of steroids. However, clinical syndromes with severe functional impairment (including swallowing disorders, myocarditis, and involvement of respiratory or laryngeal muscles, among others), and a major increase in CPK and/or myasthenia features, warrant definite ICI discontinuation and more aggressive treatment, including high-dose steroids, plasma exchange, intravenous immunoglobulin (IVIG), and immunosuppressants [70,71,72,73].

## 5. Limitations of the Review

The authors encountered difficulty in finding all relevant articles on this topic due to their multiplicity. The article therefore has the character of a narrative review, not a systematic review of the literature. Another limitation is the lack of prospective studies given the nature of complications and their rare occurrence, forcing reliance on case reports. Insufficient reporting of adverse drug reactions is also a relevant issue.

## 6. Conclusions

Clinical and histological features of main drug-related myopathies are summarized in Figure 2, Figure 3 and Figure 4.

In conclusion, muscle-related side effects are a common adverse effect of some drug groups. The pharmaceuticals described above do not represent an exhaustive list of the drugs that can cause myopathy, but the authors have tried to present the most common and relevant groups with an emphasis on the new class of medications. The purpose of this review is to enlighten the reader about the wide range of medications that can cause myopathy and the need to keep an open mind when approaching a patient with newly developed myopathy who is being treated with other medications for any underlying disease. In such cases, a detailed medication/treatment history should be obtained and any recent changes duly noted. As diagnosis may be challenging, collaboration between specialties is crucial for early detection and treatment of drug-induced myopathies. Early recognition and discontinuation of the offending drug may prevent severe adverse effects such as rhabdomyolysis and mortality. Further research is needed to better understand the pathogenetic mechanisms leading to the development of these diseases to decrease the risk of complications and provide effective treatments when they occur. Immune checkpoint inhibitors are a new group of drugs of increasing importance in oncology and, therefore, their side effects should be closely monitored to ensure the safety of treatment.

## Figures and Tables

**Figure 1 biomedicines-12-00987-f001:**
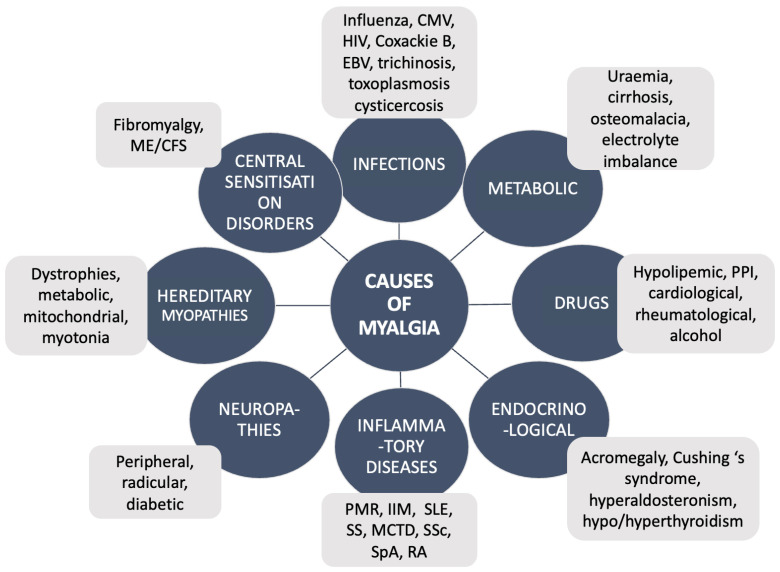
Examples of the causes of muscle pain. ME/CFS, myalgic encephalomyelitis/chronic fatigue syndrome; PMR, polymyalgia rheumatica; IIM, idiopathic inflammatory myopathies; SLE, systemic lupus erythematosus; SS, Sjogren syndrome; MCTD, mixed connective tissue disease; SSc, systemic sclerosis; SpA, spondyloarthritis; RA, rheumatoid arthritis; CMV, cytomegalovirus; HIV, human immunodeficiency virus; EBV, Eppstein-Barr virus; PPI, proton pump inhibitors.

**Figure 2 biomedicines-12-00987-f002:**
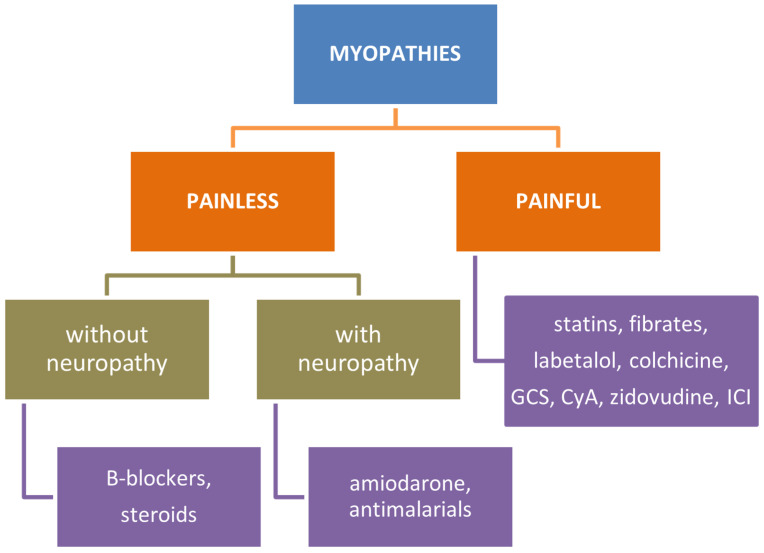
Drug-related myopathies (DRMs) distinguished by their main clinical features. GCS, glicocorticosteroids; CyA, cyclosporin A; ICI, immune checkpoint inhibitors.

**Figure 3 biomedicines-12-00987-f003:**
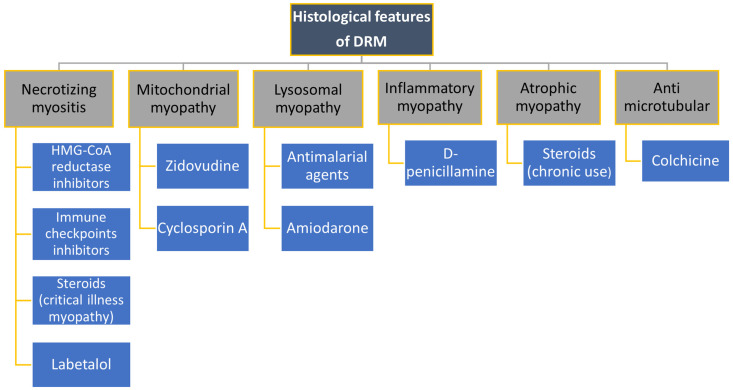
Drug-related myopathies (DRM) distinguished by their main histological features.

**Figure 4 biomedicines-12-00987-f004:**
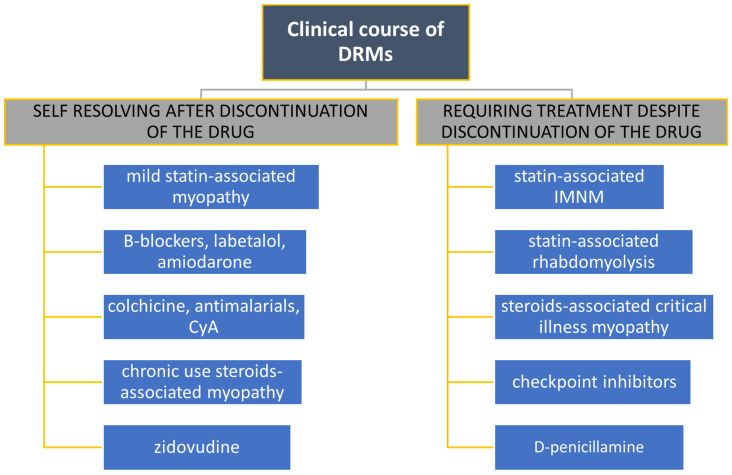
Drug-related myopathies distinguished by their clinical course.

**Table 1 biomedicines-12-00987-t001:** Diagnostic and therapeutic management in patients with myalgia.

DIAGNOSING	**Anamnesis:** age, gender, onset, location, extent, triggering and relieving factors of pain, associated symptoms, past medical history, medications ↓ **Physical examination:** fever, body mass loss, decreased muscle strength, muscle atrophy or hypertrophy, fasciculations, myotonia, asymmetrical sensory loss, “tender points”, signs of arthritis, skin lesions ↓ **Laboratory tests:** ESR, CRP, blood count, ionogram, CK, AST, ALT, creatinine, LDH, TSH, cortisol, ANA with specification, RF, ACPA ↓ **Additional examinations:** X-ray of joints, MRI of muscles, electromyography, muscle biopsy, genetic tests↓ **Multi-specialist cooperation is often necessary**	
IDENTIFICATION OF ETIOLOGY (see Figure 1)	INFECTIONS DRUGS METABOLIC ENDOCRINOLOGICAL	INFLAMMATORY DISEASES NEUROPATHIES HEREDITARY MYOPATHIES PSYCHOSOMATIC
MANAGEMENT	Depends on etiology and cause: if self-limiting → observation cause-specific treatment (e.g., antibiotics, immunosuppressive agents, hormones, antidepressants) discontinuation of potentially causative medications symptomatic treatment (warm compresses, rest, analgesics, NSAIDs, myorelaxants, physiotherapy)	

ESR, erythrocyte sedimentation rate; CRP, C-reactive protein; CK, creatinine kinase, AST, aspartate aminotransferase; ALT, alanine aminotransferase; LDH, lactate dehydrogenase; TSH, thyroid-stimulating hormone; ANA, antinuclear antibodies; RF, rheumatoid factor; ACPA, anti-citrullinated peptide antibody; MRI, magnetic resonance imaging; NSAIDs, nonsteroidal anti-inflammatory drugs.

## Data Availability

The raw data supporting the conclusions of this article will be made available by the authors on request.

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
