# Peer review of "Drug-Induced Myopathies: A Comprehensive Review and Update"

_biomedicines, 2024, doi:10.3390/biomedicines12050987_

Round 1

Reviewer 1 Report

Comments and Suggestions for Authors

Dear Authors!

Useful manuscript from the practical point of view. The frequency of myopathy is not so rare and differential diagnosis required. 

I have several concerns

1) Fig. 1 please add genetic test to the diagnostics

2) The manuscript is about myopathy, not myalgia. The Authors started from myalgia and fig 1 is about myalgia. Please explain discrepancy

3) 1st type of corticosteroid- induced myopathy is described with limitation. The clinical phenotype and information is scarse and should be elaborated.

4) Authors can create a table with short description, diagnostics, treatment and references of every type of DRM

The figures are fine and simple

The reference list corresponds to the manuscript

Comments on the Quality of English Language

Minor changes required

Author Response

Response to the comments:

Thank you very much for your suggestions. We have revised the manuscript according to your recommendations

1) we added genetic tests for the diagnostics

2) We wanted to start the text by explaining the term myalgia and to point out that it may be a frequent first symptom of myopathy, requiring proper differential diagnosis. We added also a definition of myopathy to not confuse these two terms. 

3) We elaborated the clinical illness associated - steroids myopathy. 

4) We find it very hard to create a unique table summarising all the clinical features, diagnosis, treatment and references for every type of DRM and because of that we created 3 separate figures in the last sections to summarise the article and present graphically the most important issues. 

Reviewer 2 Report

Comments and Suggestions for Authors

Study entitled "Drug-Induced Myopathies: A Comprehensive Review and Update", which presents the possible adverse effects of some drugs on muscle tissue. 

The authors use the term myopathy and myalgia interchangeably. Please review this crucial issue in your study, as it could significantly change the wording of the manuscript.

Introduction: The authors do not explain the relevance and interest of the study. It is important that the authors point out the impact of these sequelae on the muscle, as well as their influence on the quality of life of individuals. We suggest that the authors restructure the Introduction in order to present the knowledge gap.

The manuscript does not indicate objectives, material and method applied in the literature review. There is also no results section or discussion. Please, I recommend the authors to restructure the manuscript following the scientific method, even if it is a narrative review study.

In a scientific study, the summary section does not usually appear after the Introduction section. Please review the structure of the manuscript.

The figures have a small font size and colours that make them difficult to read. Please, we suggest modifying the design.

Comments on the Quality of English Language

The study is well written, needing some grammatical and spelling corrections.

Author Response

Thank you very much for your suggestions. We have revised the manuscript according to your recommendations

We wanted to start the text by explaining the term myalgia and to point out that it may be a frequent first symptom of myopathy, requiring proper differential diagnosis. We added also a definition of myopathy to not confuse these two terms. 

2. We explain the relevance and purpose of the study

3. We added to the manuscript sections: objectives, methods and limitations of the study. Discussion was performed in the section "conclusions".   

4. We modified the figures by changing the colours and enlarging the font size.

Reviewer 3 Report

Comments and Suggestions for Authors

This is an interesting manuscript though there are some gaps in the coverage of drug-induced myopathies.  Some of these gaps were mentioned in Table 2 but not discussed in the text of the manuscript and would be important additions.  The presentation of statin-induced myopathy may have included extraneous details such as grading scales.  This section could be shortened slightly and made clearer.  Other sections could also do with some clarifications.  The figures also need some corrections or clarifications.

1.      Introduction:

a.      Figure 1.  I suggest that the “Psychosomatic” bubble  that contains Fibromyalgia, ME/CFS, and depression be renamed.   There are a number of people who would object to that particular characterization of psychosomatic disorder.  Some terms to consider are “Central sensitization disorders, interceptive disorder, or medical unexplained symptoms/medically unexplained illness.

2.      Drugs Causing Myopathies

a.      Hypolipemic drugs – Could the authors clarify if necrotizing myositis related to HMGCoA antibodies is related to all of the statin related myopathies or only a subset.

b.      Cardiological drugs – Labetalol should be discussed together with the other beta blockers

c.      Colchicine – no comment

d.      Steroids – Steroid myopathy is generally thought to be related to steroid intake or Cushing’s disease.  Critical illness myopathy is considered to be associated with high dose steroids in an intensive care setting and not just from taking steroids.  Therefore I would suggest placing critical illness myopathy in a separate section from steroid myopathy.  The pathology of CIM is much more complicated than being treated with steroids.

e.      Antimalarials - no comment

f.       Cyclosporin – no comment

g.      Zidovudine – no comment

h.      Checkpoint inhibitors – The authors have not distinguished between the ICI induced myositis and ICI myasthenia.  Muscle pain, elevated CKs, and proximal weakness are predominantly ICI myositis findings.  Diplopia, dysphonia, and diplopia would suggest the presence ICI myasthenia.  The two can coexist, so the readers should be alerted to this possibility.  If there is more than muscle pain and limb weakness,  AchR levels and electrodiagnostic testing including RNS and SFEMG should be strongly considered.   In my experience, steroids are the first line of therapy for ICI myositis.  It may be the IVIG and PLEX are considered for cases suggestive of ICI myasthenia.

3.      Summary  - For figures 2-4 should chose a different color pattern than white print on yellow.  It is difficult to read on paper.

a.      Figure 2 – no comment

b.      Figure 3 – This table could be improved.  A reference that may help with this table is Neurol Clin. 2014 Aug;32(3):647-70. 

                                                    i.     Some immediate suggestions is that Necrotizing myositis – should not include steroid myopathy.  It may be some critical illness myopathies. Also, checkpoint inhibitors can have inflammatory dermatomyositis pathology or necrotizing pathology so should be included in both columns.   Labetalol could be included in this category based on the 1981 article.  Other necrotizing drug-induced myopathies include snake venom which is in the table but not discussed and propofol.  The last box on organophosphate poisoning also was not discussed in the paper and should be.  Organophosphate poisoning would be associated with over-activation of the neuromuscular junction and secondary muscle injury so may have different pathway than some other drugs.

                                                   ii.     Colchicine is considered to have antimicrotubular myopathy not lysosomal. 

                                                  iii.      D-penicillamine does cause an inflammatory myopathy as do several other medications but was not discussed in the paper.

c.      Figure 3.  ICI myositis resolves after removal of drug.  In column, requiring treatment after drug discontinuation, could the author clarify the difference between statin-associated IMNM and statin-associated rhabdomyolysis.  Are different degrees on a necrotizing myopathy.  Also, would rename steroid-associated necrotizing myopathy to include the common term of critical illness myopathy.

Comments on the Quality of English Language

There are some organizational problems with the writing but not specifically poor language.

Author Response

Thank you very much for your suggestions. We have revised the manuscript according to your recommendations

  1. Introduction - we corrected the Figure 1. 
  2. Hypolipemic drugs – we highlighted the distinctiveness of necrotizing myositis related to HMGCoA antibodies from other forms of statin- related myopathy
  3. We included labetalol in beta-blockers section
  4. We modified the section related to steroids associated myopathy
  5. Checkpoint inhibitors – We added informations about ICI- myasthenia
  6. Figures 2-4 were revisioned
  7. Figure 3 - we wanted to distinguish drug-induced myopathies by their histological image; we changed the name and included steroids-associated critical illness myopathy in necrotizing myositis (as its histological image corresponds with necrotizing myositis),
  8. We left the ICI in the field of necrotizing myopathies (their histological image corresponds with it) despite its clinical features that may remind dermatomyositis.
  9. Based on the mentioned article from 1981 we included labetalol in the category "lysosomal myopathy" as its histological image is familiar with antimalarial induced myopathy.
  10. We decided not to include the characterization of organophosphate poisoning because they are not formally drugs, we deleted them also from the table
  11.  From the histological point of view colchicine-associated myopathy has an image of lysosomal myopathy, but of course in electron microscopy  has an image of antimicrotubular myopathy
  12. Description of D-penicillamine-associated myopathy was included in the text
  13. According to the literature available and also our personal experience ICI myositis does not always resolve after removal of the drug, it may require treatment with anti-inflammatory drugs. 
  14. IMNM is characterized by an inflammatory reaction with different clinical features and a specific histopathological imagen with present anti-HMGCR-antibodies, requiring immunosupresive treatment and persistent besides discontinuation of the drug. Statin-associated rhabdomyolysis is associated with direct effects of the drug, without the inflammatory component and it does not require immunosupresive treatment.

Round 2

Reviewer 1 Report

Comments and Suggestions for Authors

Dear Authors! Thank you for your revised version of the manuscript.

I has become better and I have no any concerns

Author Response

Dear Reviewer, 

thank you once again for your suggestions on the first review and appreciation of our adjustments.

Reviewer 2 Report

Comments and Suggestions for Authors

The authors present a revised version of the manuscript, although they do not resolve point by point all the aspects that were requested in the first revision. In the introduction, new ideas have been included in one paragraph (page 2, line 36-52), although it does not resolve the events that motivated the authors to carry out the present study, and there are no solid arguments with previous studies that clarify the knowledge gap, as we suggested in the first review.

The objectives and methods sections are included, however, they do not specifically clarify the main objective of the study. Similarly, the methods section does not include inclusion criteria to facilitate the search and selection of studies. They do not indicate exclusion criteria either.

This could be related to the limitation of the study indicated by the authors.

Figures 2, 3 and 4 have been included after the conclusion section. Why?

Finally, there are no conclusions of the study.

Author Response

Dear Reviewer,

thank you for your comments.

We did our best to revise the article. We included the reasons why we chose to write this article, our main purpose was to present up-to-date information about the most relevant drug-induced myopathies. We wanted to attract the attention of clinicians to this problem and comprehensively present the topic. Our aim was also to present recently discovered immune checkpoint inhibitors-induced myopathy. In this version of the manuscript, we made further changes to explain it better.

We also corrected the objectives and methods sections but as we clarified in the limitations section we encountered difficulties in finding appropriate searching criteria to establish relevant articles due to their multiplicity. We were not able to perform a systematic review of the literature.  

We included Figures 2, 3, and 4 in the conclusion section because we think that they summarize our article and present information in an accessible way. In the present version of the article, we revised the section of conclusions to present them better.

Reviewer 3 Report

Comments and Suggestions for Authors

The revisions have covered most of my original comments.  I still request changes to Figure 3.  

Figure 3.  Colchicine is not a lysosomal disorder but an antimicrotubular myopathy so should be placed in separate column.  The labetalol articles are old but only refer to a necrotizing myopathy so should this should be moved to the first column.

Author Response

Dear reviewer, 

thank you for your comments and appreciation of our modifications.

In present version of the manuscript we revised also Figure 3. according to your suggestions.